# Noninvasive Early Detection of Nutrient Deficiencies in Greenhouse-Grown Industrial Hemp Using Hyperspectral Imaging

Alireza Sanaeifar [1], Ce Yang [1,*], An Min [1], Colin R. Jones [2], Thomas E. Michaels [2], Quinton J. Krueger [3], Robert Barnes [3] and Toby J. Velte [3]

1. Department of Bioproducts and Biosystems Engineering, University of Minnesota, Saint Paul, MN 55108, USA; sanae003@umn.edu (A.S.); mina@umn.edu (A.M.)
2. Department of Horticultural Science, University of Minnesota, 1970 Folwell Ave, Saint Paul, MN 55108, USA; jone2339@umn.edu (C.R.J.); michaels@umn.edu (T.E.M.)
3. Verilytix Inc., 2975 Klondike Avenue N, Lake Elmo, MN 55042, USA; quinton@verilytix.com (Q.J.K.); rob@verilytix.com (R.B.); toby@verilytix.com (T.J.V.)
* Correspondence: ceyang@umn.edu

**Abstract:** Hyperspectral imaging is an emerging non-invasive technology with potential for early nutrient stress detection in plants prior to visible symptoms. This study evaluated hyperspectral imaging for early identification of nitrogen, phosphorus, and potassium (NPK) deficiencies across three greenhouse-grown industrial hemp plant cultivars (*Cannabis sativa* L.). Visible and near-infrared spectral data (380–1022 nm) were acquired from hemp samples subjected to controlled NPK stresses at multiple developmental timepoints using a benchtop hyperspectral camera. Robust principal component analysis was developed for effective screening of spectral outliers. Partial least squares discriminant analysis (PLS-DA) and support vector machines (SVM) were developed and optimized to classify nutrient deficiencies using key wavelengths selected by variable importance in projection (VIP) and interval partial least squares (iPLS). The 16-wavelength iPLS-C-SVM model achieved the highest precision of 0.75 to 1 on the test dataset. Key wavelengths for effective nutrient deficiency detection spanned the visible range, underscoring the hyperspectral imaging sensitivity to early changes in leaf pigment levels prior to any visible symptom development. The emergence of wavelengths related to chlorophyll, carotenoid, and anthocyanin absorption as optimal for classification, highlights the technology's capacity to detect subtle impending biochemical perturbations linked to emerging deficiencies. Identifying stress at this pre-visual stage could provide hemp producers with timely corrective action to mitigate losses in crop quality and yields.

**Keywords:** chemometrics; hyperspectral imaging; industrial hemp; nutrient deficiencies; pre-visual detection; variable selection

## 1. Introduction

Recently, hemp (scientifically termed *Cannabis sativa* L.) has been attracting substantial worldwide interest, with a particular focus on the United States. It is interesting to see the transformation of hemp from its historical roots to a modern agricultural powerhouse, given its diverse uses across industries [1]. In addition to its use in fiber and paper production, hemp is an environmentally-sustainable and profitable crop that aligns well with eco-friendly farming practices [2]. The resurgence of industrial hemp has created an urgent need for extensive research on production management, particularly on clarifying the complex fertility requirements of cultivars bred specifically for phytochemical production. Ideal hemp fertilization depends on many factors—the specific variant grown (whether for seeds, fiber or cannabidiol (CBD)), soil characteristics, prevailing weather conditions, and achieving the right balance of essential macro- and micronutrients [3]. Determining the

optimal nutrient management strategy for hemp remains a challenging task for farmers. When levels of key elements like nitrogen, phosphorus, and potassium drop, it can really devastate yields and profits, making them constantly vulnerable. Research suggests the customizing of fertilizer regimens tailored to expected harvests, based on proven techniques dialed in for other major crops. The soil conditions and microclimate of each operation as well as other factors must be considered when optimizing a specific hemp operation [4].

Deficiencies could also leave the plants more exposed when new pathogens emerge, potentially catalyzing disastrous impacts. Identifying the mechanisms inducing these effects early and deploying strategic protections may help mitigate the carnage [5]. Meticulously monitoring plant health is also critical, especially in less than stellar environments, to enable proper maturation. Therefore, really comprehensive crop surveillance remains vital, facilitating accurate preemptive yield loss forecasts and proactive preparedness planning [6]. Furthermore, among the many complex facets of precision agriculture that need advanced methods, one really stands out—developing new plant varieties and picking the best ones for specific conditions. Therefore, there is a pressing need for fast, precise, non-invasive phenotyping techniques to effectively evaluate plant stress levels and enhance breeding programs [7].

In recent years, there has been growing interest in using state-of-the-art imaging technologies to measure how abiotic stress impacts plants. Adopting non-destructive imaging allows continuous measurements over time, enabling ongoing monitoring of crop resilience under stress [8]. An exciting new technology called hyperspectral sensing has recently gained popularity, offering a way to holistically assess plant physiology by seamlessly combining spatial and spectral information. It represents one of several cutting-edge techniques that have the potential to revolutionize our understanding of plant reactions and adaptation to environmental stressors [6]. The unique spectral signatures of leaves and canopies can potentially be tapped as insightful markers of shifts in plant health due to changing conditions. Each plant pixel holds a wealth of intricate details on its chemical composition and status, allowing a comprehensive appraisal of its health. Both mature and young leaves show the effects of nutrient deficiencies, ultimately restricting growth. These shortfalls initially become visible under light as pigment changes, with leaf yellowing producing higher reflectance in the green–red wavelengths [9]. Studies demonstrate that necrotic areas linked to deficiencies exhibit increased reflectance in the near-infrared (NIR) and shortwave infrared (SWIR) regions, whereas areas with non-necrotic nutrient deficiencies often show decreased reflectance in those spectral regions. Therefore, hyperspectral imaging is emerging as a promising non-invasive tool for assessing crop nutrition and tracking stress changes over time [10].

Daily tracking of fluctuations in plant physiology is key to grasping subtle hourly shifts and impacts. Weksler et al. [11] developed a mobile hyperspectral camera to continuously capture a large amount of greenhouse measurements throughout the day. By syncing the spectral data with sensors in real-time, they gained insight into the dynamic interplay between spectral signatures and pepper plant responses to potassium. Specific spectral bands were significantly correlated with momentary water loss rates. This shows the power of hyperspectral imaging to monitor plant behavior in real-time. In a groundbreaking study, Siedliska et al. [12] built models to track phosphorus levels during different growth phases of various crops fertilized in different ways. Leveraging hyperspectral imaging and machine learning, they successfully sorted plants into four categories based on phosphorus fertilization levels. Their discoveries unveiled the capacity of combining hyperspectral imaging and machine learning for precise quantification of phosphorus at an early stage in the growth cycle. Moreover, a substantial enhancement in classification accuracy was observed as plants progressed through successive growth stages. This really spotlights the power of the fusion of imaging and machine learning to grasp phosphorus dynamics in crops and manage them effectively. In another study, Osco et al. [13] used hyperspectral analysis on Valencia orange leaves, developing machine learning algorithms to quantify macro- and micronutrients like nitrogen, phosphorus, potassium, magnesium, sulfur,

copper, iron, manganese, and zinc. Random forest models showed the highest accuracy and predictive ability compared to other models. The Relief-F algorithm was key in identifying the most important wavelengths for nutrient prediction. The impressive results in estimating key nutrients highlighted the effectiveness and robustness of their approach for assessing nutrients in Valencia orange leaves.

While hyperspectral imaging has shown promise for physiology and stress assessment in various crops, there has been limited but growing interest in using it for monitoring industrial hemp. Several recent studies demonstrated the effectiveness of hyperspectral imaging techniques in the near-infrared and shortwave infrared spectral regions for differentiating hemp cultivars, growth stages, and plant components like stems, leaves, and flowers [14–16]. Machine learning approaches attained high classification accuracies up to 100% for factors like plant component and growth stage. Additionally, this technology was combined with chemometrics for non-destructive measurement of key cannabinoids in industrial hemp flowers and leaves [17]. However, no published studies have leveraged spectral imaging specifically for early diagnosis of nutrient deficiencies in industrial hemp prior to symptom manifestation.

In the cultivation of industrial hemp within greenhouse environments, the prompt identification and resolution of deficiencies in crucial nutrients such as nitrogen, phosphorus, and potassium (NPK) hold paramount significance. Our research goal was to find out if NPK deficiencies could be identified early, before symptoms arise. The aim was to give growers tools to detect NPK issues way before visible signs show up, so that they can take action to maintain adequate levels of these essential nutrients in greenhouse hemp plants. Our initial emphasis involved the identification of optimal wavelengths for precise detection of nitrogen, phosphorus, and potassium (NPK) deficiencies in plants. Subsequently, a meticulous selection of the appropriate algorithm was made to identify these deficiencies at an early stage in the growth cycle. The provision of such capabilities to growers could significantly impact their productivity.

## 2. Materials and Methods

### 2.1. Experimental Greenhouse Setup and Hemp Cultivar Selection

This experiment was carried out in the controlled greenhouse space at the University of Minnesota's Plant Growth Facilities on the St. Paul campus. Three hemp cultivars and breeding lines—Trilogene Alpha, Atlas Wilhelmina, and UMN 5-4—representing various plant phenotypes were selected in this study. Trilogene Alpha and Atlas Wilhelmina are specialized feminized auto-flowering types carefully bred for CBD production, while UMN 5-4 is a hardy breeding line from the feral hemp collections that the University of Minnesota has nurtured along. In the first step, healthy seedlings were transplanted into 3-gallon pots around 10 days after planting (DAP) with one plant per pot. The pots were loaded with Promix BRK growing media with a modest dose of Osmocote Plus 15-9-12 fertilizer (3–4 months) at 3 g per liter. The purpose of this low fertilizer rate was not to feed these young plants completely—it was simply to maintain them until they reached maturity. It was not our intention to produce hemp at a commercial level.

Around 50 DAP, two treatments were introduced to the experiment. The first group of healthy plants received Jack's 20-10-20 Peat-Lite nutrient at 150 ppm using a careful bottom-water fertigation method. This continued throughout and served as our control group (CK). The second group, labeled nutrient deficient (ND), did not receive any additional nutrients beyond the initial Osmocote dose. The experiment consisted of pots, with each pot containing one plant from each of the three hemp cultivars (Trilogene Alpha, Atlas Wilhelmina, UMN 5-4). A total of 54 pots were included in the experiment (9 pots per cultivar × 2 nutrient treatments × 3 cultivars). Six pots per cultivar/treatment group were randomly chosen for detailed photography and analysis. The whole experimental setup was replicated twice. Conducting the replication gave us robust data collection and analysis.

## 2.2. Hyperspectral Data Acquisition and Preprocessing

Spectral measurements at three critical time points in the study were obtained—T1 (55 DAP), T2 (60 DAP), and T3 (64 DAP) in a dark room (Figure 1b). The main goal was to detect stress early, ideally before any visible symptoms appeared. The dark room conditions reduced ambient light interference/variability to enable reliable early stress detection through stabilized imaging [18]. All the selected samples underwent careful scanning with a calibrated Pika L 2.4 (Resonon Inc., Bozeman, MT, USA) benchtop hyperspectral imaging system. A 23 mm lens was mounted on the camera, providing a field of view of 15.3 degrees. The camera has a spectral resolution of 2.1 nm. The camera is a line-scan camera, also called push-broom imager. The system is comprised of a linear stage assembly propelled by a motor. Overhead, we carefully positioned halogen lighting to optimize the lighting condition for the image collection. The system operated through Spectronon Pro 3.4.8 software (Resonon Inc., Bozeman, MT, USA), which was connected to the camera via a USB cable. Before scanning any plants, a series of key calibration steps were conducted. Dark correction was conducted by blocking the lens and capturing multiple dark frames. Then camera calibration was conducted using both the dark image and an image captured from the white Spectrolon reference panel.

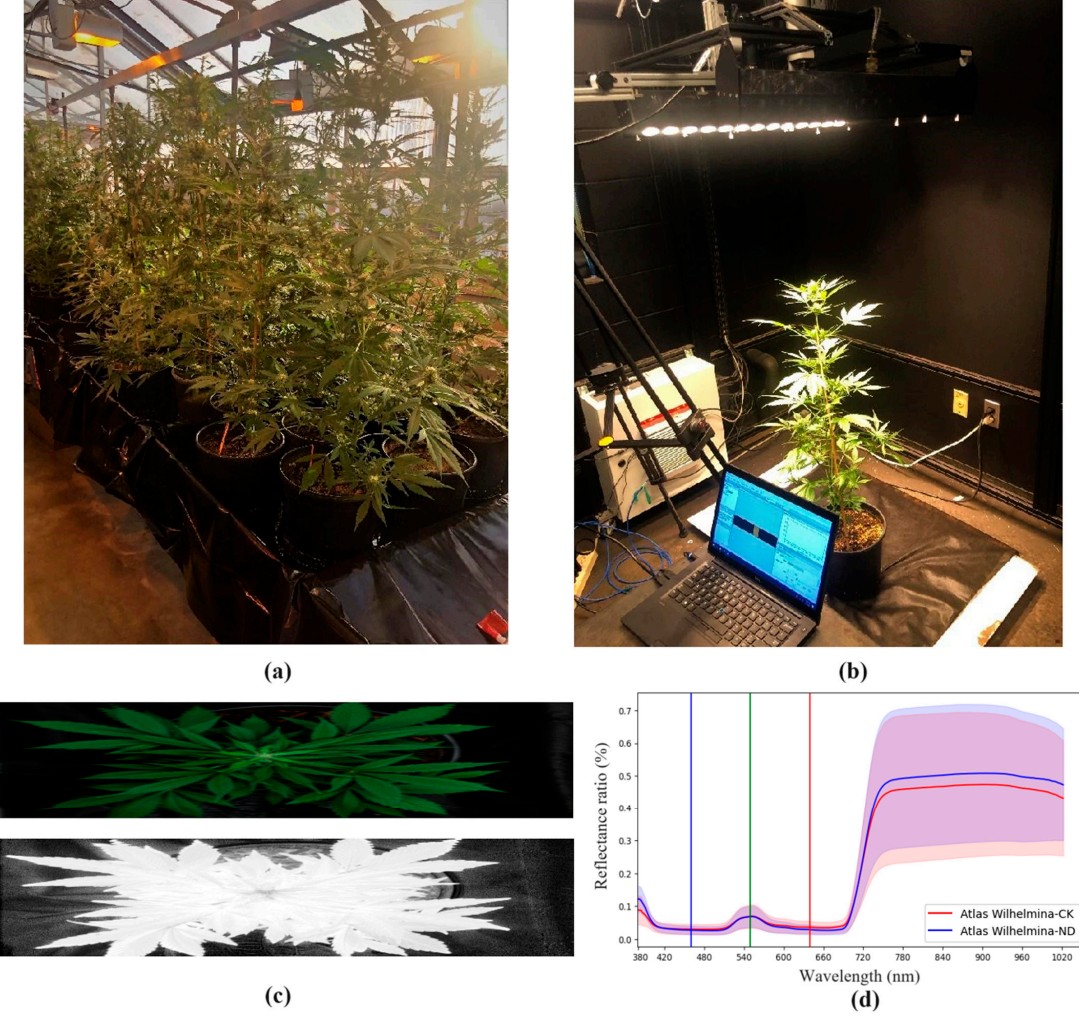

**Figure 1.** (**a**) Industrial hemp plants in the greenhouse; (**b**) setup of spectral measurement equipment for hemp plants using a hyperspectral camera and halogen lighting; (**c**) NDVI analysis to distinguish hemp plants from the background; (**d**) average spectral signatures of control and nutrient-deficient Atlas Wilhelmina hemp plants (sample size = 108), with shaded areas indicating standard deviation (SD = 0.0125–0.226).

To enable precise data extraction and analysis, we relied on the widely used normalized difference vegetation index (NDVI) (Figure 1c). This helps distinguish the crop plants from the background [19]. We then carefully picked three adjacent regions within each plant that had similar spectral profiles. The selection process involved using Euclidean distance measurements to pinpoint those zones of interest (ROIs). We aimed to have each ROI cover around 2000 pixels, capturing representative leaf tissue—both symptomatic and asymptomatic areas—to collect comprehensive data across the entire leaf. For each control and nutrient-deficient group at different time points, we examined 6 plants total, choosing 3 distinct regions within each plant. This resulted in 18 samples collected from each group. Table 1 shows the breakdown of the number of samples used for data processing and analysis.

**Table 1.** Number of samples used for data processing and analysis.

| Variety/Nutrient Deficiency Stage | T1 | T2 | T3 |
| --- | --- | --- | --- |
| Atlas Wilhelmina | CK (18), ND (18) | CK (18), ND (18) | CK (18), ND (18) |
| Trilogene Alpha | CK (18), ND (18) | CK (18), ND (18) | CK (18), ND (18) |
| UMN 5-4 | CK (18), ND (18) | CK (18), ND (18) | CK (18), ND (18) |

After each scanning session, we utilized the Spectronon Pro for pre-processing to extract the spectral data from the leaf samples, hoping to gain some useful insights to advance our research goals. Potential disruptions, including equipment noise, quirks in experimental methods, environmental factors, and stray light, can impair data analysis accuracy and lead to less reliable findings. Therefore, spectral data require careful pre-processing prior before being analyzed. For this work, we thoroughly compared different pre-processing techniques before selecting the Sklearn MinMaxScaler as our first step. This method is particularly suitable for scaling the pixel values for each spectral band to a consistent 0–1 range. Then we applied group scaling, relying on the notion that the magnitude of a measurement reflects its importance, with relatively uniform noise across variables. Group scaling means dividing the variables into evenly-sized blocks, then scaling each block by the grand mean of their standard deviations. Within each block, this approach calculates the deviations of the variables, then uses their averages to standardize all the columns in that block. The process needs to be repeated for each block of variables, assuming equal column numbers as a default. This ultimately yielded harmonized, noise-reduced data ready for multivariate analysis [20].

### 2.3. Multivariate Analysis of Hyperspectral Data

2.3.1. Optimizing Spectral Analysis through Robust Outlier Detection and Noise Reduction

For this research endeavor, we allocated 66% of the 324 samples (shown in Table 1) to the calibration set and 34% to the prediction set using the Kennard–Stone algorithm—a method that provides coverage across sets for creating reliable models [20]. To further promote stability and dependability in our models, which can vary in this field, we conducted 10-fold Venetian blind cross-validation. By creating 10 distinct splits with a blind thickness of one sample, overfitting was prevented [21]. We find this strengthens confidence in model evaluations. Additionally, we omitted 10 wavelengths from both ends of the original 300 wavelength range. This left us with 280 wavelengths between 400.71 to 999.6 nm to utilize for downstream analysis. From our empirical experiences, we discerned that noise frequently manifests at the edges of the spectral range, potentially introducing distortion into the analysis.

In analyzing our dataset of 324 samples, we leveraged robust principal component analysis (RPCA) to excise potential outliers and boost the accuracy of subsequent data analysis. RPCA has proven to be uniquely beneficial when handling noisy or irregular data that may skew results if not addressed. As mentioned above, we removed 10 wavelengths at both ends of the spectrum under examination to reduce the impact of noise. Through RPCA, we could decompose the data into its low-rank skeleton capturing the intrinsic structure

and a sparse component isolating deviant outliers and noise [22]. This decomposition enabled pinpointing and discarding of aberrant values that diverged markedly from the norm and would have detrimentally skewed the analysis. While dimension reduction techniques always risk discarding meaningful signals, our judicious component selection minimized this likelihood. Further simulation studies and real-world trials are needed to fully probe the impacts of RPCA-based preprocessing on downstream statistical testing.

### 2.3.2. Wavelength Selection

Hyperspectral data, characterized by their multidimensional nature and inherent redundancies, have the propensity to impede modeling performances. Therefore, applying variable selection algorithms is key to pinpointing the optimal wavelengths and boost model performance. The right algorithm can simplify things by cutting redundant data and axing irrelevant variables. For this work, we evaluated two algorithms—variable importance in projection (VIP) and interval partial least squares (iPLS)—to identify the most useful wavelengths. VIP scores estimate the importance of each variable in the PLS projection. Variables with VIP scores near or above one are considered influential, while variables with scores significantly below one are less important and may be excluded. This method provided an initial filtering to isolate the spectral regions with the strongest signal [23]. iPLS selects a subset of variables that provides superior prediction compared to using all variables. It does an exhaustive search for the optimal variable or combination. iPLS can operate in "forward" mode, with intervals successively included, or "reverse" mode with intervals successively removed. In this study, we used iPLS in forward mode starting with individual PLS models for each defined variable interval [24]. Cross-validation was performed on each model and the interval giving the lowest root mean square error of cross-validation (RMSECV) was selected first. RMSECV refers to the standard deviation of the differences between predicted values and actually observed values in a CV setting. This continued until the specified number of intervals was reached or RMSECV plateaued. The interval size used in this study was one variable. We compared the performance of VIP and iPLS in selecting the key wavelengths carrying information on early nutrient stress indicators. This enabled the assessment of which approach was optimal for reducing data volume while retaining predictive power—crucial for developing an accurate and rapid detection system.

### 2.3.3. Development of Classification Models

The spectral data, originally stored as .xls files, were imported into MATLAB R2021b (MathWorks, Natick, MA, USA) for analysis. We then used the PLS_Toolbox version 8.9.1 (Eigenvector Research Inc., Manson, WA, USA) to train classification models with robust features. We employed partial least squares-discriminant analysis (PLS-DA) to classify the nutrient deficiency stages. PLS-DA is a supervised technique, so it requires labeled data to establish connections between spectral features and predefined classes, such as the nutrient deficiencies here [25,26]. The goal was to build a solid supervised model to discriminate the samples across three varieties regarding control (CK) and different deficiency stages (T1, T2, T3). Specifically, PLS-DA finds latent variables, also called PLS components, that maximize the relationship between the predictor variables (spectra) and the response (deficiency stage). These components project the data into a lower dimensional space while preserving key discriminatory information, enabling effective classification. The model learns to differentiate the stages based on this condensed spectral representation, allowing accurate categorization into the specified classes. This approach empowers effective discrimination of nutrient stress even with the complexity of hyperspectral data.

In addition to PLS-DA, we also explored support vector machines (SVMs) for developing classification models. SVMs are nonlinear machine learning models well-suited for classification tasks. The key idea behind SVMs is to project the data into a higher dimensional space using kernel functions and construct an optimal separating hyperplane between classes in that space [20,27]. The kernels effectively transform the data to make it

more linearly separable. The hyperplane is positioned to maximize the margin between classes, which improves generalization performance. Specifically, the SVM model consists of a subset of training samples known as support vectors, as well as nonlinear kernel coefficients that together define the transformation and hyperplane [28]. The model enables prediction of the class membership for new samples based on their mapped positions relative to the hyperplane. We utilized the svm function in MATLAB which implements SVMs using the LIBSVM package. LIBSVM provides two commonly used SVM formulations—C-SVM and nu-SVM. C-SVM uses a regularization parameter C to control the penalty applied to misclassified training samples. Nu-SVM uses a 'nu' parameter to specify the maximum fraction of allowed training errors and support vectors. Both solve the same optimization problem. A key advantage of the SVM function is that it automatically tunes the C, nu, and kernel gamma parameters using cross-validated grid search. This avoids the need for manual parameter tuning. However, single values can be specified if desired [29]. Once optimized, the SVM model discriminates between classes using the nonlinear spectral project. We applied SVM using the key wavelengths selected by VIP and iPLS as predictors. The nonlinear SVM models provided an alternative approach to PLS-DA for classification of nutrient deficiency stages across varieties and time points. We compared the performance of optimized SVM and PLS-DA models to determine the optimal technique.

The performance of the developed PLS-DA and SVM classification models was thoroughly evaluated using several key metrics:

- Precision—The proportion of correctly classified positive samples out of all positive classifications. Higher values indicate greater effectiveness.
- Sensitivity—Measures the model's ability to correctly identify positive cases. A sensitivity of 1 means all positive cases are detected.
- Specificity—Evaluates how well the model identifies negative cases. A specificity of 1 means no false positives occur.
- F1 Score—The harmonic mean of precision and sensitivity. Provides overall measure of model accuracy accounting for both false positives and false negatives.
- Class Error—The proportion of misclassified samples out of all samples. Lower values are better.

These metrics were calculated using a confusion matrix, which summarizes the predictions of the model. The confusion matrix is an N-by-N table contrasting the true and predicted classes. The diagonal cells show correct classifications (true positives/negatives) while the off-diagonal cells show errors (false positives/negatives). From the confusion matrix, the sensitivity, specificity, and class error are calculated using the counts of true/false positives and negatives. Overall precision is simply the sum of correct positive predictions divided by total positive predictions. In addition to these quantitative metrics, we generated classification images visually depicting the distribution of correctly and incorrectly classified samples. An effective model will demonstrate high accuracy, sensitivity, and specificity along with low class error. By analyzing the multiple metrics, we evaluated the ability of PLS-DA and SVM models to identify nutrient stress in hemp plants.

## 3. Results and Discussion

### 3.1. Spectral Response of Hemp Samples under Nutritional Stress

The reflectance and absorption of light are strongly influenced by the physiological and chemical characteristics of plants, which can change under stress and impact the reflectance spectrum. The near-infrared and visible ranges are crucial for plant growth measurements. Using these wavelengths enables hyperspectral imaging to detect changes in leaf pigment (400–700 nm) and cellular composition (700–1300 nm) [30].

Since chlorophyll aids photosynthesis and absorbs light, fluctuations in chlorophyll due to stress can alter light interactions. Stressed plants may show depleted chlorophyll detected as low 530–630 nm reflectance and increased 700 nm reflectance. Beyond chlorophyll, pigments like carotenes and xanthophylls also contribute to light reflection [31]. Carotenoids and anthocyanins also help defend against environmental factors [6,32]. In

addition, physical leaf traits like tissue morphology, cell walls, and thickness may change under stress, influencing spectral properties. Stomata can also adversely impact moisture, gas exchange, and temperature, affecting infrared reflectance. Altered cell components, proteins and carbohydrates under stress significantly impact reflectance as well [33,34].

Spectral data reveal the absorption of key chemical groups like C-H, N-H, and O-H, containing composition and structure information. Nutrient-deficient samples showed similar overall spectral trends to controls but different reflectance magnitudes (Figure 1d). A 550 nm peak was seen, as hemp mostly absorbs blue-violet and red light for photosynthesis but reflects green light, giving the leaves their color. Some weak 900–1000 nm peaks were observed, attributed to -CH3 groups [35]. The peaks relate to internal leaf components. Therefore, the spectral response enables qualitative and quantitative analysis. The 555 nm peak relates mainly to chlorophyll content. The 800–1200 nm region corresponds to leaf structure, water content, and chemical components. Importantly, the spectral pattern of nutrient-deficient and control hemp leaves was similar across the wavelengths. No clear characteristic peaks related to the stress were observed. Therefore, to determine nutrient stress, multivariate data analysis is needed, working off the subtle spectral differences between samples.

### 3.2. Outlier Detection Using RPCA

When developing calibration models from spectroscopy data, it is common to find some data points that diverge from the main trends. Excluding these non-fitting outliers can improve the model's reliability [36,37]. We used two key statistics—Q-residuals and Hotelling's $T^2$—to detect outliers among 324 samples, each with 280 wavelengths. Q-residuals measure the variation unexplained by the model, calculated from the error matrix. Samples with large Q-residuals versus others demonstrate poor model fit. Hotelling's $T^2$ indicates the variation within the model, based on the score values. Lower scores reflect better fit. Therefore, Hotelling's $T^2$ represents the distance from an ideal perfect fit [38]. We first computed the Q-residuals and Hotelling's $T^2$ using a PCA model's scores and loadings, and then identified points outside the 95% confidence range for both metrics, flagging 44 outliers. In general, samples with big Q-residuals are poorly modeled, whereas samples with high Hotelling's $T^2$ values greatly vary within the model. After removing these 44 outliers, there were 280 samples remaining for further analysis. Using Q-residuals and Hotelling's $T^2$ combined allowed for the successful screening of anomalous points and the improvement of the calibration data. Figure 2 shows the Hotelling $T^2$, PC1, and PC2 values versus the Q residuals obtained for the RPCA model.

### 3.3. Temporal Classification of Nutrient Deficient Stress in Hemp Plants Using PLS-DA

Prior to investigating classification under stress conditions, a preliminary analysis was conducted to assess the efficacy of our methodology in discerning variations in the visible and near-infrared spectra among diverse hemp cultivars. Accordingly, a PLS-DA model was constructed, wherein each cultivar class could be distinctively segregated based solely on the control group data. The model had three regression vectors (and predicted Y values), one for each of the three cultivar classes—Atlas Wilhelmina-CK, Trilogene Alpha-CK, and UMN 5-4-CK. As seen in Figure 3a, the predicted Y values for the control samples nicely classified the three cultivars. Based on Bayes' theorem, a threshold (represented by the red line) was determined to minimize errors. Figure 3b shows VIP scores indicating each wavelength's importance for PLS projection. Variables with VIP near or above one are influential, while those far below one are less useful and could be excluded. We found wavelengths around 700 nm were best for distinguishing the hemp cultivars.

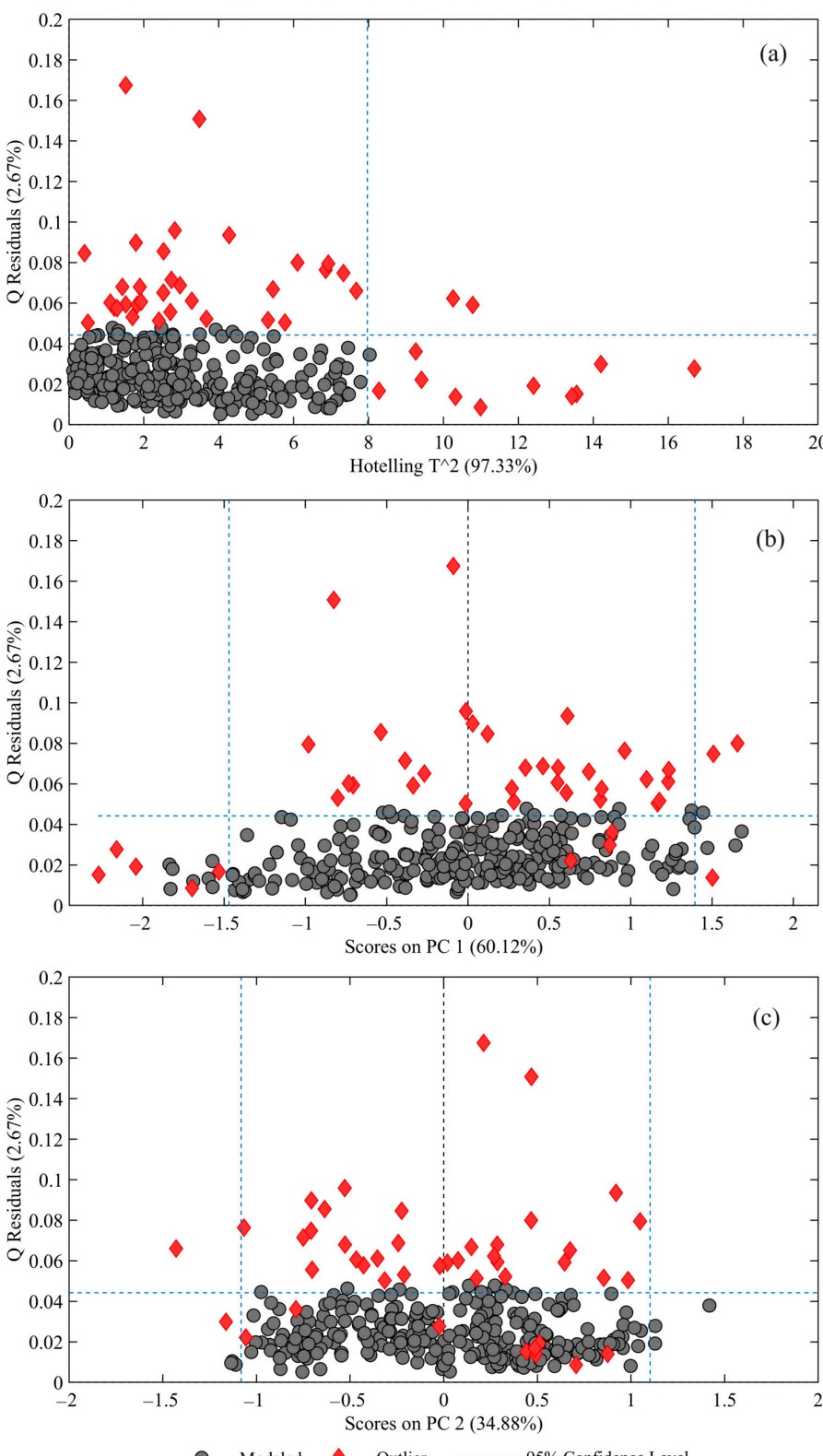

**Figure 2.** RPCA model analysis for outlier detection. (**a**) Hotelling T2, (**b**) principal component 1 (PC1) and (**c**) principal component 2 (PC2) scores versus Q residual scatter plot. Samples categorized as outliers (exceeding 95% confidence level, blue dashed line) are indicated in red, while retained samples are colored gray.

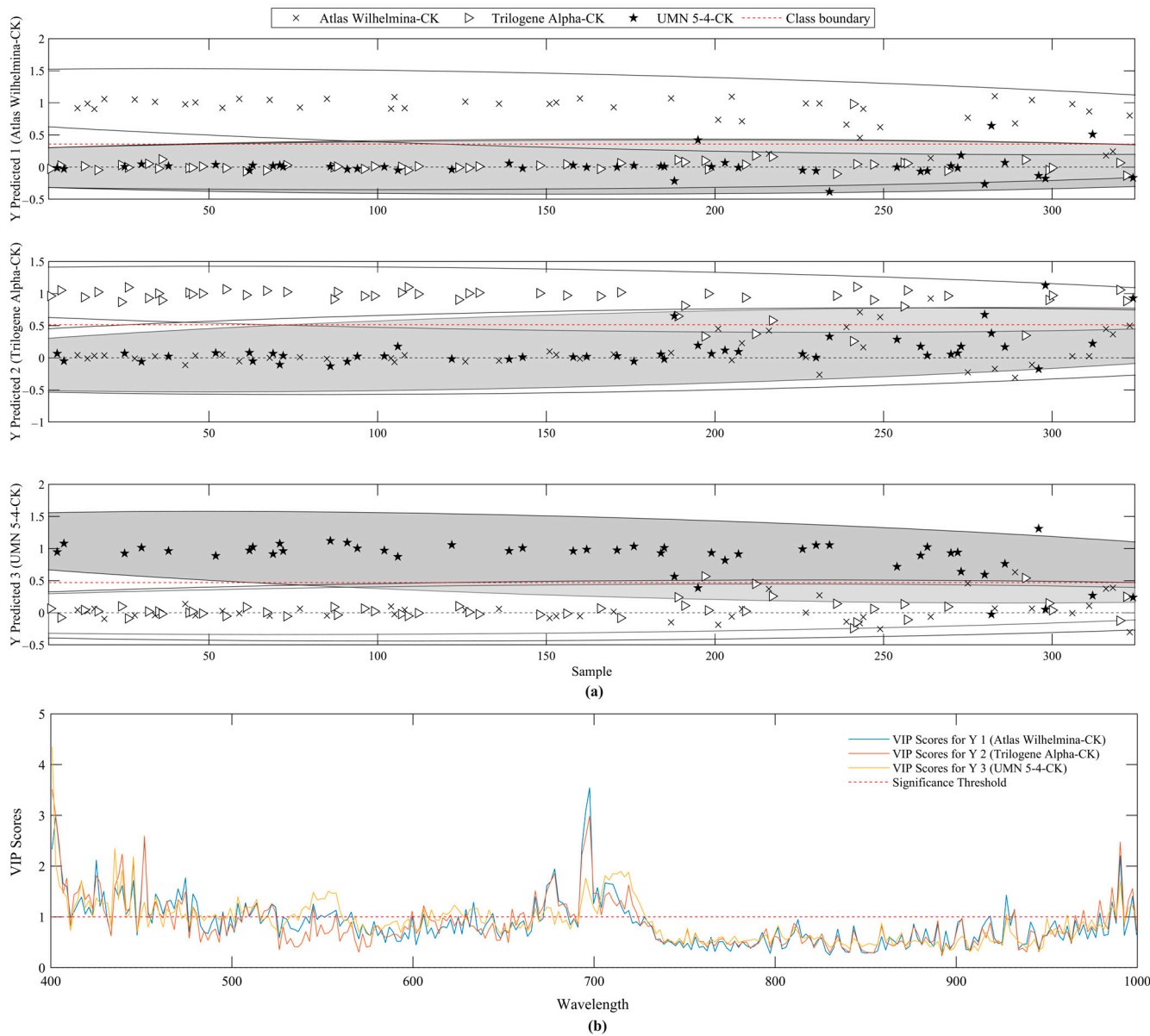

**Figure 3.** (**a**) Predicted Y values from the PLS-DA model to distinguish between the three hemp cultivars. (**b**) VIP scores indicating the importance of wavelengths for cultivar discrimination.

After first building the PLS-DA model on the control groups, we proceeded and applied it to detect early nutritional stress. The model included classes for both the control (CK) and nutrient-deficient groups at three different time points (T1-ND, T2-ND, T3-ND) across the three hemp varieties. As shown in Figure 4a, the model performed well in classifying the various nutrient-deficient groups of plants, even with multiple cultivars included. In other words, this implies its potential for generalizable stress detection beyond individual cultivars.

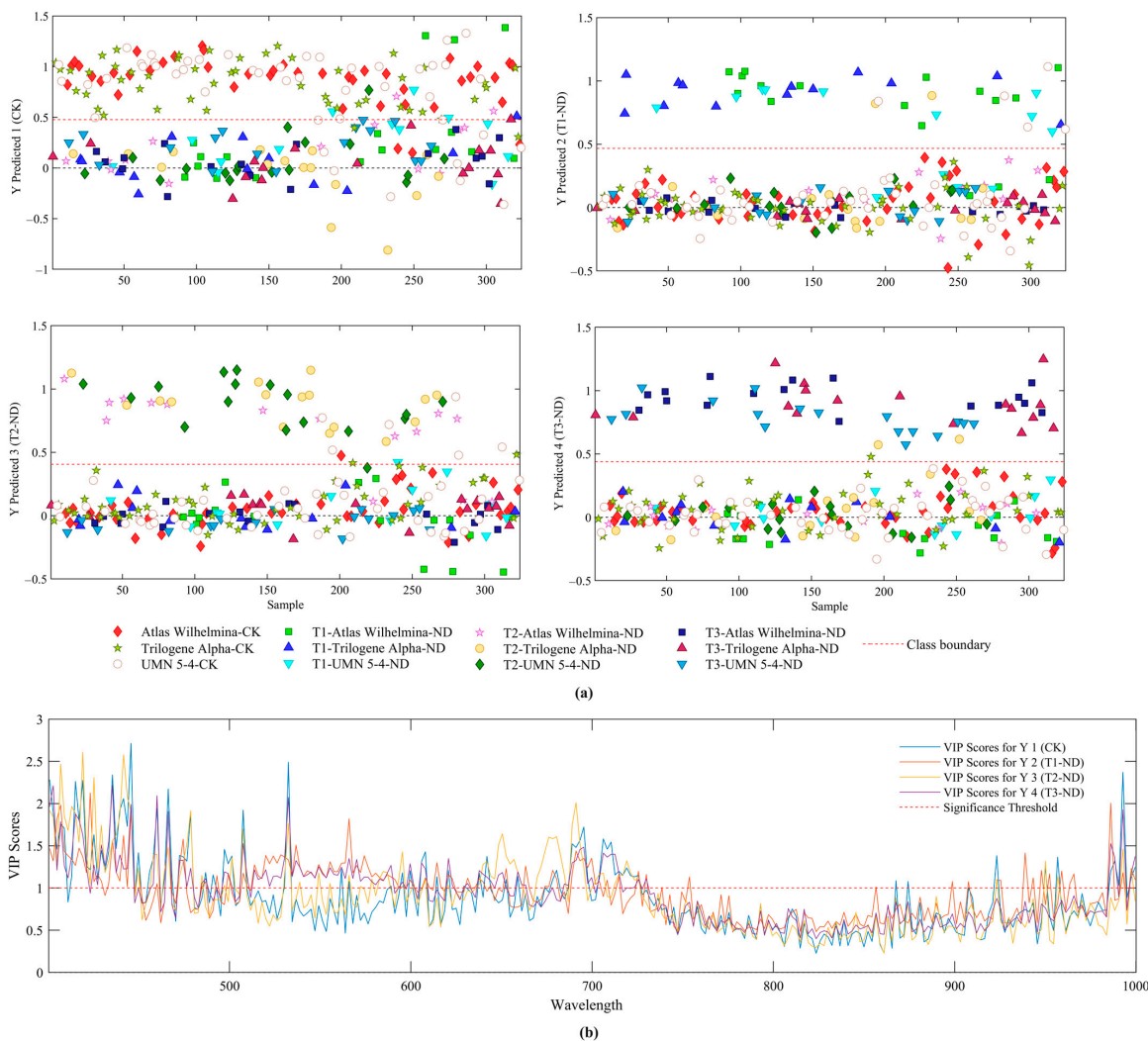

**Figure 4.** (**a**) Predicted Y values from the PLS-DA model for early detection of nutrient stress across control and deficient groups. (**b**) VIP scores indicating the most important wavelengths for discriminating nutrient deficiency groups.

Comparing the VIP scores in Figures 3b and 4b, we noticed the most useful wavelengths for distinguishing between control samples shifted over to the green–red region of the visible spectrum when it came to distinguishing the nutrient deficiency groups. This is consistent with previous findings that nutritional deficiencies often affect the visible range the most due to pigmentation changes and result in higher reflectance in the green–red region at wavelengths close to 540 nm and 650 nm [6,9]. Figure 4b highlights the importance of the green and red parts of the spectrum for detecting nutritional deficiencies in hemp, similar to what has been observed in other plants. Table 2 presents the performance metrics of the PLS-DA model for classifying nutrient deficiency at different time points (T1-ND, T2-ND, T3-ND) compared to the control (CK) across the training, validation, and test data. For the training set, the model achieved strong performance with sensitivity, specificity, and precision all equal to 1.0. On the validation set, sensitivity ranged from 0.76 to 0.96, specificity from 0.85 to 0.99, precision from 0.85 to 0.89, and F1 scores from 0.80 to 0.92, indicating robust model performance on unseen data. Comparable results on the test set confirm the model's ability to detect reliably early nutrient stress. While T1 deficiencies were most difficult to detect with 0.67 test sensitivity, T3 stress was perfectly classified with 1.0 test sensitivity. Overall, PLS-DA demonstrated reliable discrimination of nutritional deficiencies in hemp using hyperspectral imaging. Further enhancement may focus on improving sensitivity to subtle early T1 deficiencies.

**Table 2.** Performance metrics for temporal classification of nutrient deficient stress in hemp plants using the PLS-DA model.

| Class | | Sensitivity | Specificity | Class Error | Precision | F1 |
|---|---|---|---|---|---|---|
| CK | Train | 1.000 | 1.000 | 0.000 | 1.000 | 1.000 |
| | Validation | 0.883 | 0.848 | 0.135 | 0.850 | 0.866 |
| | Test | 0.787 | 0.841 | 0.185 | 0.828 | 0.807 |
| T1-ND | Train | 1.000 | 1.000 | 0.000 | 1.000 | 1.000 |
| | Validation | 0.773 | 0.985 | 0.045 | 0.895 | 0.829 |
| | Test | 0.667 | 0.932 | 0.113 | 0.667 | 0.667 |
| T2-ND | Train | 1.000 | 1.000 | 0.000 | 1.000 | 1.000 |
| | Validation | 0.759 | 0.969 | 0.071 | 0.846 | 0.800 |
| | Test | 0.684 | 0.924 | 0.113 | 0.619 | 0.650 |
| T3-ND | Train | 1.000 | 1.000 | 0.000 | 1.000 | 1.000 |
| | Validation | 0.964 | 0.969 | 0.032 | 0.871 | 0.915 |
| | Test | 1.000 | 0.990 | 0.008 | 0.958 | 0.979 |

Receiver operating characteristic (ROC) curves were used to visualize the specificity and sensitivity trade-offs across different classification threshold settings for the PLS-DA model predictions (Figure 5). Specificity and sensitivity refer to the percentages of correctly classified negative and positive cases below and above a given threshold, respectively, as illustrated in Figure 4a. The area under the ROC curve (AUC) effectively summarizes model discrimination ability by integrating specificity and sensitivity across thresholds. The AUC values exceeded 0.81 for all deficiency classes and control, signifying the PLS-DA model performed successfully in temporally classifying nutrient stress in hemp using hyperspectral imaging.

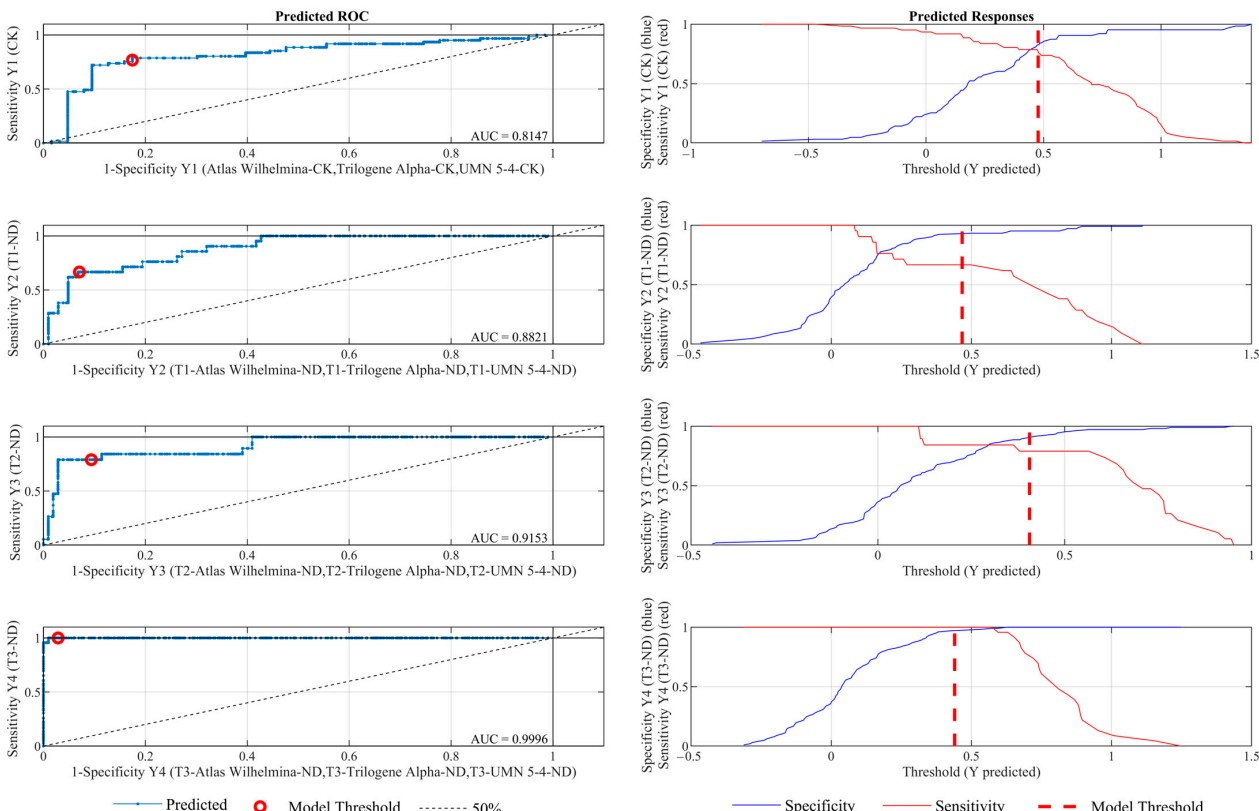

**Figure 5.** ROC curve and threshold analysis for temporal classification of nutrient stress in hemp plants.

### 3.4. Enhancing Classification Performance Using SVM Models Combined with Wavelength Selection

Before creating the SVM models, we applied VIP and iPLS wavelength selection methods to extract the most significant wavelengths for improving and streamlining spectral analysis [7,39]. The original data contained 280 wavelengths, with many redundant and co-correlated variables. These irrelevant wavelengths not only negatively affect model performance but also increase calculation time. Using appropriate variable selection to identify key influential wavelengths, along with an advanced classifier, could benefit nutrient stress classification in hemp. iPLS gave the greatest dimensionality reduction, from 280 down to 16 wavelengths. The method works by constructing PLS models iteratively, discarding wavelengths with the smallest regression coefficients each time to isolate useful predictors [20]. VIP reduced the number of wavelengths down to 70 by ranking them according to their significance to the PLS model projection.

With the optimal wavelengths identified, we evaluated different SVM configurations to improve classification over PLS-DA. We developed and compared optimized C-SVM and nu-SVM models using the radial basis kernel function (RBF) to handle nonlinear relationships. Crucially, SVM requires optimizing the cost, nu, and kernel gamma parameters. We used 10-fold cross-validation grid search on the training set to determine the values giving minimal error. To build a robust model, SVM parameters were carefully tuned across a wide range. The kernel gamma shapes the separating hyperplane and indicates the number of variables. The cost balances errors and complexity. Nu sets bounds on support vectors and training errors [40,41].

In Table 3, we show how optimized SVM models utilizing key wavelengths selected by VIP and iPLS can distinguish nutrient deficiencies over time in hemp from healthy controls. We tested the reduced wavelength subsets with C-SVM and nu-SVM architectures. Across all nutrient stress classes, the 16-wavelength iPLS-C-SVM model achieved 0.75 to 1 precision on the test set, which outperformed the other architectures. The iPLS-C-SVM also demonstrated notable sensitivity and specificity of over 0.79 for each class. These results suggest that combining robust variable selection and SVM classification could prove useful for early deficiency identification from complex spectral data. Moreover, the SVM models exhibited superior performance compared to the 280-variable PLS-DA model. Particularly, the utilization of the streamlined iPLS wavelength set in conjunction with optimized C-SVM parameters yielded the most favorable outcomes. This approach retained essential discriminatory information while substantially reducing the overall size of the data. Figure 6 shows the optimal combination of SVM parameters yielding the minimum misclassification error for the iPLS-C-SVM and iPLS-Nu-SVM models. For these architectures, the lowest error occurred with cost = 100, gamma = 0.031623, and nu = 0.11. Tuning these hyper-parameters is critical for proper SVM model configuration and generalization for accurate classification of nutrient stress from the hyperspectral data. This visualization reveals valuable perspective into how modifying model complexity and margin thresholds can impact performance. Overall, the optimized settings produced robust SVM models with strong discrimination ability for early deficiency detection in hemp using a selective subset of predictive wavelengths. Table 4 demonstrates the confusion matrix for the top-performing iPLS-C-SVM model on the test set. This matrix shows the actual vs. predicted nutrient deficiency classes, and the diagonal cells show samples that were correctly classified. Additionally, there are off-diagonal elements, which are errors. For the control group, the model correctly classified 52 of 61 samples as CK, with five misclassified as T1-ND and four as T2-ND. For T1-ND, 18 of 21 were correctly predicted, with confusion mainly associated with the CK class. The T2-ND and T3-ND classes had good performance, with 15/19 and 19/23 correct predictions respectively. There were no samples left unassigned, and some difficulty distinguishing early T1 deficiencies from healthy controls could be an area for improvement in the future. Overall, the model did a good job discriminating, with low misclassifications. For the iPLS-C-SVM model, we also present the predicted class probabilities for the different hemp cultivars across each of

the four nutrient deficiency categories in Figure 7. This allows for a better visualization of the cultivars according to their nutritional deficiencies. The ability to classify early-stage nutrient deficiencies accurately makes this model ideal for plant health monitoring. In this study, we defined the T1-ND class as samples imaged only 4 days after stress application. The fact that the optimized iPLS-C-SVM model could correctly classify 18 out of 21 T1-ND samples highlights the technique's sensitivity to early changes. Being able to detect deficiencies at such an early stage, before visible symptoms manifest, provides an invaluable window for growers to proactively apply corrective treatments and prevent further damage. It is likely that these early biochemical changes would have been missed if the plants had only been assessed visually. However, hyperspectral imaging can detect these changes through subtle pigment fluctuations. Overall, the capacity to classify stressed plants after just four nutrient-deficient days underscores the vast potential of hyperspectral imaging to shift practices from reactive to preventative assessment, intervention and management in hemp and other high-value crops.

**Table 3.** Results of SVM models for temporal classification of nutrient deficient stress in hemp plants using optimal wavelengths.

| Wavelength Selection Method | Number of Wavelengths | SVM Type | SVM Optimal Parameters | Class | Sensitivity | Specificity | Class Error | Precision | F1 |
|---|---|---|---|---|---|---|---|---|---|
| VIP | 70 | C-SVM | Cost = 100 Gamma = 0.00031623 | CK | 0.836 | 0.778 | 0.194 | 0.785 | 0.810 |
| | | | | T1-ND | 0.571 | 0.942 | 0.121 | 0.667 | 0.615 |
| | | | | T2-ND | 0.684 | 0.952 | 0.089 | 0.722 | 0.703 |
| | | | | T3-ND | 1.000 | 1.000 | 0.000 | 1.000 | 1.000 |
| | | Nu-SVM | Nu = 0.275 Gamma = 0.0001 | CK | 0.852 | 0.778 | 0.185 | 0.788 | 0.819 |
| | | | | T1-ND | 0.571 | 0.961 | 0.105 | 0.750 | 0.649 |
| | | | | T2-ND | 0.737 | 0.952 | 0.081 | 0.737 | 0.737 |
| | | | | T3-ND | 1.000 | 1.000 | 0.000 | 1.000 | 1.000 |
| iPLS | 16 | C-SVM | Cost = 100 Gamma = 0.031623 | CK | 0.852 | 0.841 | 0.153 | 0.839 | 0.846 |
| | | | | T1-ND | 0.857 | 0.951 | 0.065 | 0.783 | 0.818 |
| | | | | T2-ND | 0.789 | 0.952 | 0.073 | 0.750 | 0.769 |
| | | | | T3-ND | 0.826 | 1.000 | 0.032 | 1.000 | 0.905 |
| | | Nu-SVM | Nu = 0.11 Gamma = 0.031623 | CK | 0.820 | 0.841 | 0.169 | 0.833 | 0.826 |
| | | | | T1-ND | 0.857 | 0.942 | 0.073 | 0.750 | 0.800 |
| | | | | T2-ND | 0.789 | 0.943 | 0.081 | 0.714 | 0.750 |
| | | | | T3-ND | 0.826 | 1.000 | 0.032 | 1.000 | 0.905 |

**Table 4.** Confusion matrix for the iPLS-C-SVM model on the test set.

| | | Actual Class | | | |
|---|---|---|---|---|---|
| | | **CK** | **T1-ND** | **T2-ND** | **T3-ND** |
| | CK | 52 | 3 | 4 | 3 |
| | T1-ND | 5 | 18 | 0 | 0 |
| Predicted as | T2-ND | 4 | 0 | 15 | 1 |
| | T3-ND | 0 | 0 | 0 | 19 |
| | Unassigned | 0 | 0 | 0 | 0 |

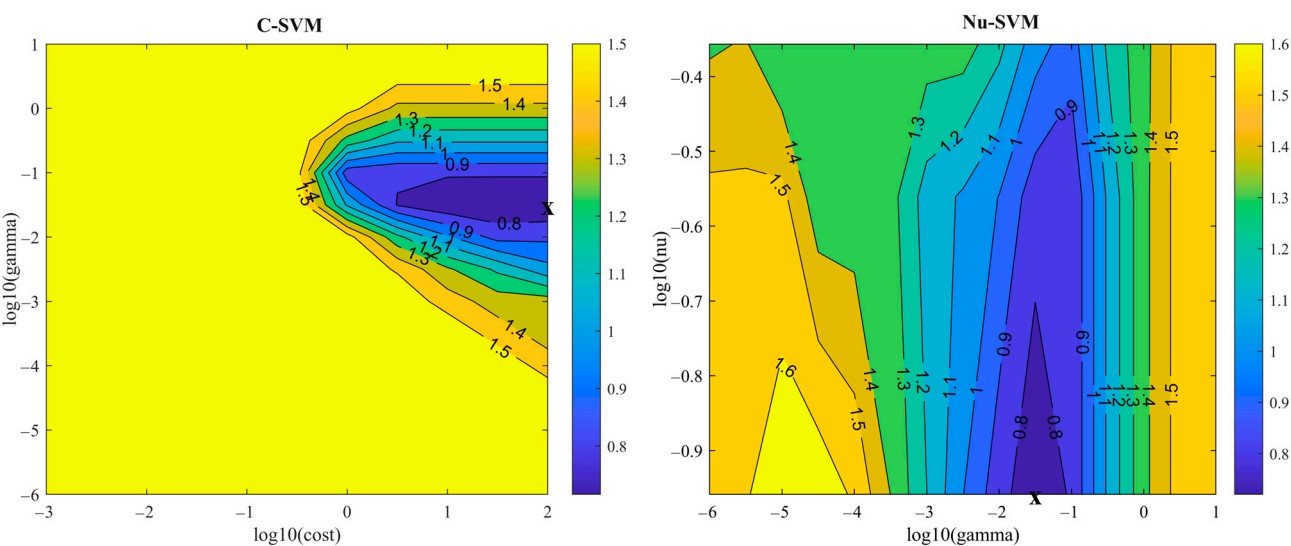

**Figure 6.** Visualization of the optimal SVM parameter configuration for the iPLS-C-SVM and iPLS-Nu-SVM models, denoted by "X", that minimized misclassification errors.

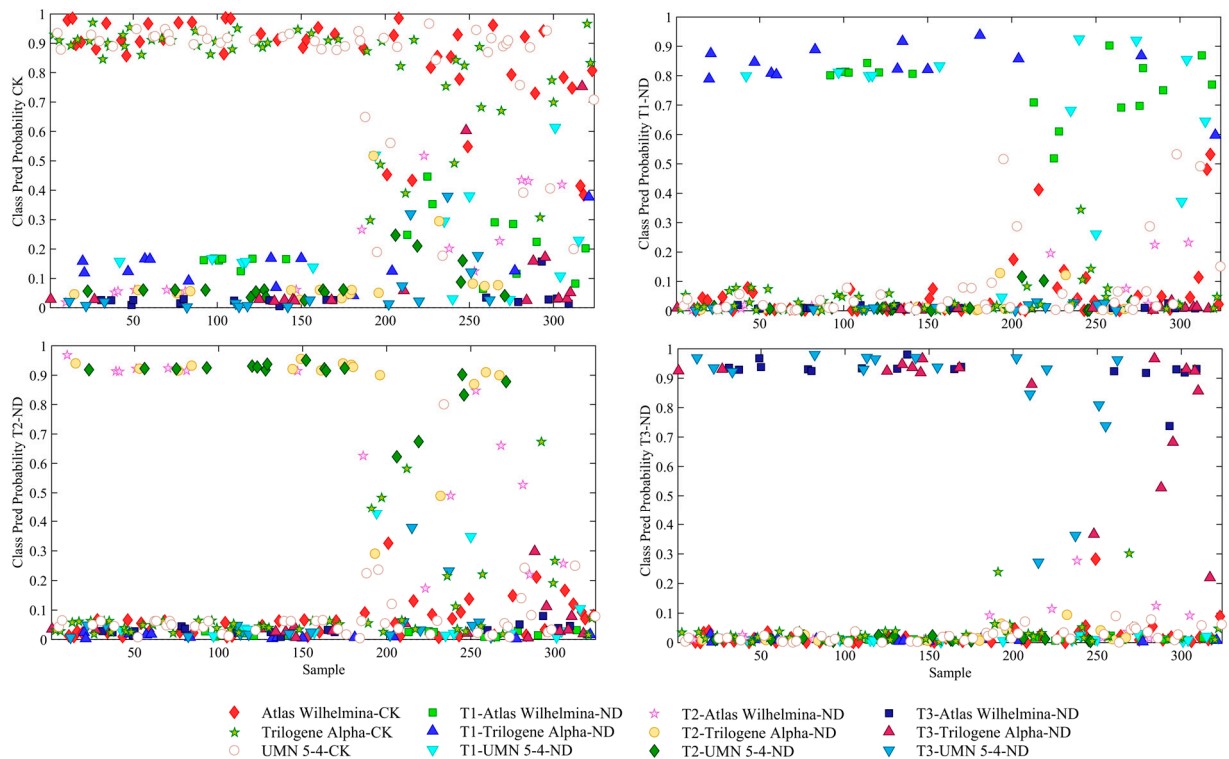

**Figure 7.** Predicted class probabilities for different hemp cultivars across the four nutrient deficiency classes: control (CK), and deficient stages at time points T1, T2, and T3 (T1-ND, T2-ND, T3-ND).

We used extreme gradient boosting discriminant analysis (XGBDA) to determine the importance of each optimal wavelength for classification. XGBDA calculates variable importance by summing the reduction in loss function ("gain") at nodes where that variable was used for splitting, across all models [42]. The variables are then ranked by cumulative gain to identify the most influential wavelengths. Figure 8 revealed that the most impactful wavelengths were within the visible range, specifically highlighting 568.02 nm and 712.62 nm to be the most important wavelengths in this study. The visible range's primacy highlights its effectiveness in detecting biochemical changes linked to chlorophyll and other pigments occurring due to emerging nutrient deficiencies. Chlorophyll, the

primary pigment involved in photosynthesis, strongly absorbs blue and red visible light while reflecting green wavelengths. Carotenoids and anthocyanins are secondary pigments that also impact leaf color [6]. When plants experience nutritional stress, alterations in pigment concentrations are typically among the earliest biochemical changes, preceding declines in growth or visible symptom development [43]. For example, chlorosis caused by nitrogen deficiency induces breakdown of chlorophyll molecules, allowing more green light to reflect and causing leaves to appear more yellowed. Hyperspectral imaging in the visible range can identify these subtle pigment fluctuations before they are clearly visible by the naked eye [44]. The emergence of the visible range, particularly wavelengths related to chlorophyll absorption and reflection, as most important for classification, underscores hyperspectral imaging's sensitivity to early stress detection based on indications of impending pigment and biochemical changes.

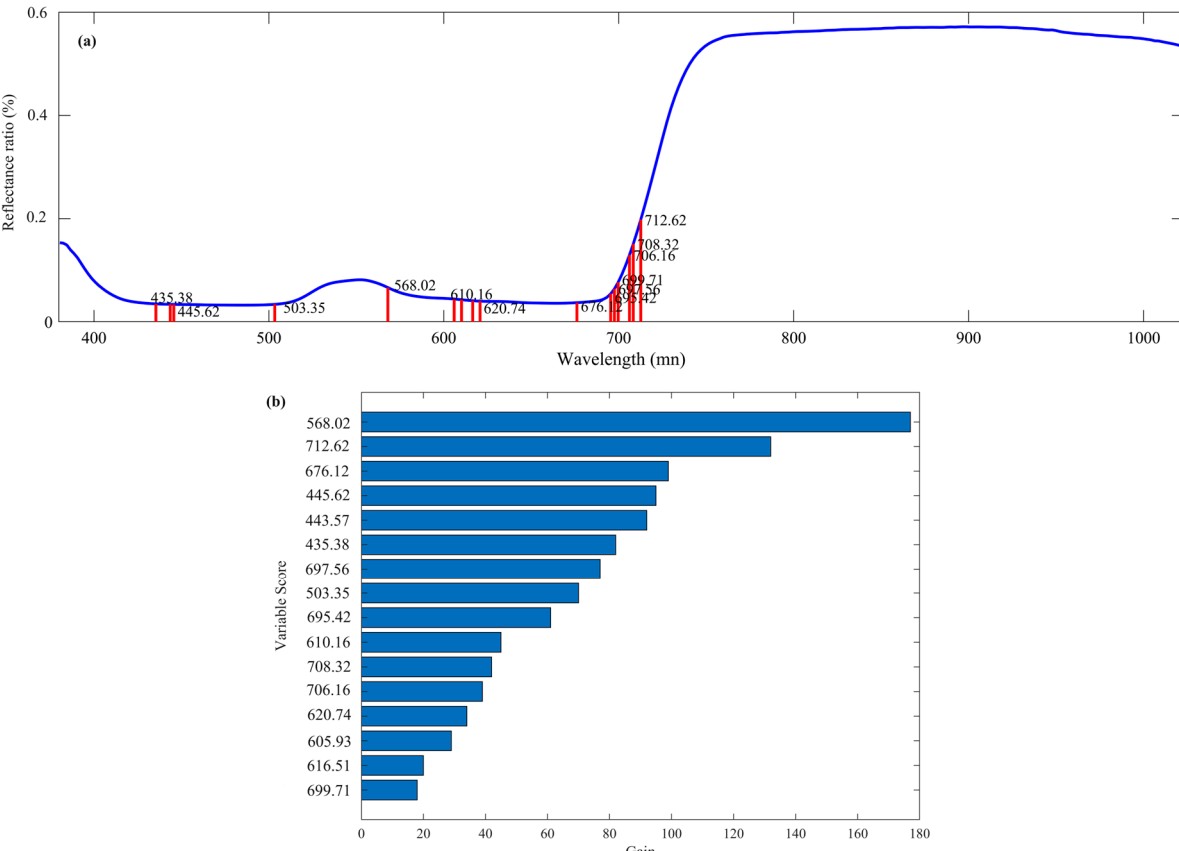

**Figure 8.** (**a**) Location and (**b**) relative importance of the 16 optimal wavelengths identified for nutrient deficiency detection using iPLS analysis.

This research explored the use of hyperspectral imaging for pre-visual diagnosis of nutritional stress in *Cannabis sativa* (hemp) grown in a controlled greenhouse setting. Initial findings demonstrated proof-of-concept feasibility; however, further validation is required before recommending adoption for widespread agricultural applications. The cultivation techniques utilized differ from common large-scale production methods in two key aspects—restricted container sizes prevented full plant maturity realization and controlled environmental conditions lacked real-world variability. As protected greenhouse cultivation of hemp increases, these initial results may inform early diagnostic approaches focused on nutrient deficiencies to support greenhouse growers. While hemp can thrive in both greenhouse and field production, additional testing under diverse commercial growth conditions will confirm whether the findings translate to field settings. Moreover, while a calibrated imaging system was utilized under optimized lighting parameters, real-world

data collection varies substantially in illumination intensity, temperature, humidity, and more. To transition these preliminary positive outcomes into robust, broadly applicable producer recommendations, expanded collaborative field trials across major cultivation zones are planned. Through partnerships with leading hemp producers, key limitations of the current proof-of-concept study can be addressed via commercial-scale, on-farm field assessment. Future efforts will prioritize developing practical solutions that consider the inevitable constraints of large-scale agricultural production. With further validation, hyperspectral imaging has the potential to serve as a rapid, non-destructive tool for crop status monitoring to support hemp producers.

## 4. Conclusions

This study aimed to evaluate the potential of hyperspectral imaging for early, non-invasive diagnosis of nutrient deficiencies in industrial hemp. Our results successfully demonstrated proof-of-concept, classifying plants based on nitrogen, phosphorus, and potassium status using spectral data alone. The iPLS-C-SVM approach offered significant advantages; by using only key wavelengths, it reduced data volume and computational needs. Thus, the initial objective of developing an effective spectroscopic technique for pre-visual nutrient deficiency screening was attained through the chemometric methods applied. Our central hypothesis that nutrient deficiencies alter leaf biochemistry prior to visible symptoms—creating discernible spectral signatures in affected plants—was also validated by the classification findings. Farmers could benefit greatly from the ability to assess crop nutrition in real time. Instead of waiting until deficiencies visibly manifest, they could identify issues early and implement corrective measures to maintain yield and quality. Further field-scale testing is needed to translate these results into on-farm recommendations. There are also opportunities to integrate aerial systems, advanced phenotyping techniques like fluorescence, and more diverse datasets to increase robustness. In closing, this proof-of-concept study shows promise for hyperspectral imaging to enable optimized nutrient management in hemp cultivation. With additional validation and method development, the approach could become part of responsible intensification strategies as the hemp industry expands. Transitioning this academic research to user-friendly tools will require collaboration between researchers, industry partners, and farmers. Overall, these findings help us move closer to in-field monitoring tools to support evidence-based decisions around supplementing nutrients for optimal hemp growth and production.

**Author Contributions:** Conceptualization, A.S. and C.Y.; methodology, A.S.; software, A.S.; validation, A.S. and C.Y.; formal analysis, A.S.; investigation, A.S. and C.Y.; resources, A.S., C.R.J. and C.Y.; data curation, A.M.; writing—original draft preparation, A.S.; writing—review and editing, C.Y.; visualization, A.S.; supervision, C.Y. and T.E.M.; project administration, C.Y.; funding acquisition, Q.J.K., R.B. and T.J.V. All authors have read and agreed to the published version of the manuscript.

**Funding:** This work was funded by Verilytix Inc. to the University of Minnesota with the funding Number CON000000095565.

**Data Availability Statement:** The data presented in this study are available on request from the corresponding author.

**Conflicts of Interest:** Q.J.K., R.B. and T.J.V. were employed by the company Verilytix Inc. The remaining authors declare no conflict of interest.

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
