# Peer review of "Noninvasive Early Detection of Nutrient Deficiencies in Greenhouse-Grown Industrial Hemp Using Hyperspectral Imaging"

_remotesensing, doi:10.3390/rs16010187_

Round 1

Reviewer 1 Report

Comments and Suggestions for Authors

This is a well-written paper presenting novel research on using hyperspectral imaging for early detection of nutrient deficiencies in industrial hemp plants. The methodology is sound, combining controlled greenhouse experiments with advanced spectral data analysis using multivariate techniques like RPCA, PLS-DA, SVM, and variable selection methods. The results clearly demonstrate the capability to classify nutrient stresses, even at an early pre-visual stage, with good accuracy. The paper makes a strong contribution towards enabling real-time crop monitoring to support timely interventions. I recommend acceptance after minor edits.

Specific comments:

  1. In the abstract, the sentence "The 16-wavelength iPLS-C-SVM model demonstrated the strongest performance, achieving precision of 0.75 to 1 on the test dataset." should be changed to "The 16-wavelength iPLS-C-SVM model achieved the highest precision of 0.75 to 1 on the test dataset."
  2. In the introduction, the sentence "Figuring out ideal nutrient management for hemp continues to be a challenging task for farmers." should be changed to "Determining the optimal nutrient management strategy for hemp remains a challenging task for farmers."

3.       Line 38: Change "in addition to conventional fiber and paper applications" to "in addition to its use in fiber and paper production"

  1. Line 67: The use of "burst onto the scene" could be replaced with a more technical phrase for consistency in the scientific context.
  2. Line 69: Consider rephrasing "coming down the pike" for a more formal tone.
  3. Line 197: Replace "lends credibility to the model assessments" with "strengthens confidence in model evaluations"

7.       Line 83: Replace "a ton" with "a large amount"

8.       Line 90: Replace "plants into four distinct phosphorus fertilization levels" with "plants into four categories based on phosphorus fertilization levels"

9.       Check for consistency in terminology, especially regarding hyperspectral imaging (HSI).

  1. Line 101: Replace "the power and reliability of their approach" with "the effectiveness and robustness of their approach"
  2. Line 137: The sentence "so there were two complete replications." is redundant and could be deleted.
  3. Line 145: Replace "A 23 mm lens was mounted on the camera with a field of view is 15.3 degrees." with "A 23 mm lens was mounted on the camera, providing a field of view of 15.3 degrees."
  4. Line 307: Replace "During plant growth measurements, the near-infrared and visible ranges play a key role." with "The near-infrared and visible ranges are crucial for plant growth measurements."

14.    Line 434: Replace "impact model performance" with "negatively affect model performance"

Reviewer 2 Report

Comments and Suggestions for Authors

Dear authors,I want to congratulate you for the work you have made and provided new knowledge for the specific issue within this article. I have only a few comments in the attached file. Please consider them or justify your decision.

Reviewer 3 Report

Comments and Suggestions for Authors

This mansucript (remotesensing-2773745) showed that hyperspectral imaging effectively detects early nutrient deficiencies in industrial hemp plants by analysing spectral data and using algorithms such as PLS-DA and SVM. This “noninvasive” technology is sensitive to changes in leaf pigments and identifies previsual nitrogen, phosphorus, and potassium stresses, aiding in timely interventions to maintain crop quality and yield.

I suggest revising the title to 'noninvasive', because 'pre-nonvisual?' is not an adequate term. This is an interesting manuscript, with robust statistical analyses, appropriate selection and application methods, as well as its relevance in the field of proximal remote sensing, applying imaging tools. Minor corrections are necessary, mainly in the legends of the figures (they should reflect as much information as possible and not be summarized, as the authors have done; see specific comments below). I also consider that the unique sections of results and discussion could be improved by adding relevant and appropriate references, which should be easily resolvable by the authors. Be careful with the concepts of absorbance or absorbance factor when referring to them in the text or figures, as some passages are incorrect. I also suggest including a topic both in the introduction and in the discussions about the potential of plants when applying other classification, prediction, and modelling tools (many of which have been published in Remote Sensing in 2023).

Keywords in alphabetical order;

Please check scientific names and alter for italicization. Ex. L103, and Table 1.

(Figure 1b) without the dot.

In the Materials and Methods section, it is necessary to add references.

The legends of the figures and tables should contain all the elements that appear and provide the most detailed description possible about abbreviations, sample size, and scale.

Figure 1: Why did not the authors truncate the x-axis at 380 nm? Why are the absorbance values high? On the y-axis, the correct terms are 'Absorbance Factor' or 'Absorbance' when represented as a percentage. However, the values presented are far from the standard. Could the authors explain or correct the figure appropriately? What was the sample size? Can you explain about the SED or SD plotted on the curves?

Section 3. Results and Discussion needs to improve the citations and include them in paragraphs where they are missing, e.g., Lines 333-348, and L431-505 as well as in other sections. Please check and correct this.

L278-291: I suggest to the authors not to write in bullet points, as this makes the text appear 'poor'. L286: The font is different in '0.99'.

Figure 2 needs an addition of an explanation for the three images, as well as an explanation about the colors of the points and lines presented. Even though there is a 'legend' within the images, these should also be described in the captions. Consider for other figures.

Figure 8 “Fator of absorbance”.

Include in the conclusions whether the initial objectives and hypothesis were indeed achieved.

Check old references.

Comments on the Quality of English Language

Minor changes in grammar and spelling.

Round 2

Reviewer 3 Report

Comments and Suggestions for Authors

The manuscript has been improved. I recommend it for publication.

Comments on the Quality of English Language

Minor corrections in gramamr.